# Quantifying the Effect of Land Use Change Model Coupling

**Oleg Stepanov** [1] , **Gilberto Câmara** [2] **and Judith A. Verstegen** [1,*]

[1] Institute for Geoinformatics, University of Münster, 48149 Münster, Germany; oleg.v.stepanov@gmail.com

[2] National Institute for Space Research (INPE), São José dos Campos 12227-010, Brazil; gilberto.camara@inpe.br

\* Correspondence: j.a.verstegen@uni-muenster.de

**Abstract:** Land-use change (LUC) is a complex process that is difficult to project. Model collaboration, an aggregate term for model harmonization, comparison and/or coupling, intends to combine the strengths of different models to improve LUC projections. Several model collaborations have been performed, but to the authors' knowledge, the effect of coupling has not been evaluated quantitatively. Therefore, for a case study of Brazil, we harmonized and coupled the partial equilibrium model GLOBIOM-Brazil and the demand-driven spatially explicit model PLUC, and then compared the coupled-model projections with those by GLOBIOM-Brazil individually. The largest differences between projections occurred in Mato Grosso and Pará, frontiers of agricultural expansion. In addition, we validated both projections for Mato Grosso using land-use maps from remote sensing images. The coupled model clearly outperformed GLOBIOM-Brazil. Reductions in the root mean squared error (RMSE) for LUC dynamics ranged from 31% to 80% and for total land use, from 10% to 57%. Only for pasture, the coupled model performed worse in total land use (RMSE 9% higher). Reasons for a better performance of the coupled model were considered to be, inter alia, the initial map, more spatially explicit information about drivers, and the path-dependence effect in the allocation through the cellular-automata approach of PLUC.

**Keywords:** land-use change; model coupling; partial equilibrium model; demand-driven model; Brazil; validation

## 1. Introduction

Land-use change (LUC) is an important direct form of human impact on the environment [1] and a key factor contributing to anthropogenic greenhouse gas (GHG) emissions [2,3]. LUC directly affects ecological, biophysical and biochemical processes and system states such as biodiversity, freshwater storage and flow regimes [4]. Furthermore, it indirectly influences climate by changing characteristics of the earth's surface such as soil moisture and albedo [5,6]. To be able to evaluate the impacts of ongoing and future LUC, it is crucial to understand spatial processes behind LUC, to which much work has been dedicated in recent decades [7,8].

Different models have been developed for projecting[1] LUC. Each of these models has its strengths and weaknesses, due to, for example, spatial scale (reflecting a specific decision-making level of actors, such as individual farmers, a group of land owners, or spatial planners), thematic application

---

[1] Whereas a prediction assumes that future changes in a system's conditions will not influence the future system state, a projection specifically accounts for changes in the conditions [9]. The sets of possible conditions are then typically captured in scenarios. As such, a weather model makes predictions but a LUC model or climate model makes projections.

boundaries (e.g., models focusing on a single land-use (LU) class vs. more general multi-class models), and timeframe constraints (e.g., general computable equilibrium models projecting for long time frames vs. partial equilibrium models for shorter time frames) [1].

The term "model collaboration" was introduced to describe how various models may be linked to each other with the aim to reduce the above-mentioned weaknesses of individual models [10]. There are three components to model collaboration that vary by degree of model interconnectivity: harmonization, comparison, and coupling. Model harmonization deals with the alignment of input data, ontologies and semantics, spatial and temporal extents and scenarios. Model comparison evaluates model parameters and methods, input data and model results. Model coupling involves either loose/soft coupling where the output of one model is used as input for the other, or tight/hard coupling where there is two-way communication allowing for feedbacks between the coupled models [10,11]. On the one hand, hard coupling makes the structure of the coupled model more complicated and less transparent, as several cycles of communication may have occurred between the two models before output is generated, but the system is described more consistently. On the other hand, in soft linking, the communication often occurs via files that can be inspected. Soft linking also provides the ability to couple more parts of the models, although this requires strict control of the possible inconsistencies in the data [11].

Model collaboration is becoming a standing practice in LUC modeling. For example, Prestele et al. [1] and Alexander et al. [12] compared a harmonized set of land-use classes across a set of global-scale LUC models to identify hotspots of disagreement between the models. Furthermore, Lapola et al. [13], Verstegen et al. [14] and Meiyappan et al. [15] have coupled a Computable General Equilibrium (CGE) model to a spatially explicit demand-driven model to study LUC impacts. In addition, Halofsky et al. [16] used model collaboration to assess the influence of climate change on local vegetation shifts. Yet, to the authors' knowledge, no study exists that assesses whether a harmonized, coupled LUC model, as presented in e.g., [14], actually performs better than an individual LUC model, when compared to independent observational data. Since model harmonization and coupling is time-consuming, we deem it important to determine the added value of such an effort. Therefore, our aim was to fill this knowledge gap by comparing the performance of a harmonized, coupled LUC model with a single LUC model.

We present a modeling setup in which we harmonized, loosely coupled and compared a bottom-up Partial Equilibrium (PE) model, GLOBIOM-Brazil [17–19], and a spatially explicit demand-driven model, the PCRaster Land Use Change model (PLUC) [14,20], for a case study of LUC in Brazil from 2007 to 2030. First, we assessed model performance for the period 2007 to 2015. Hereto, we compared the results of GLOBIOM-Brazil and the results of the coupled model with independent observational data: a time series of land-use maps up to 2015 developed by the Brazilian Space Agency (INPE) [21]. Next, we compared the projections of GLOBIOM-Brazil to the projections of the coupled model for 2030. We aimed to answer the following research questions: (1) What are the differences between land-use patterns produced by the coupled model and GLOBIOM-Brazil and what do these differences tell us about the models? (2) For which land use classes, if any, does the coupled model produce better results than GLOBIOM-Brazil individually when being validated against independent observational data?

Brazil is one of the world leaders in the production of livestock, food crops and biofuels [22]. Dias et al. [23] claimed that agriculture is the main driver of deforestation, changes in soil and water, and significant loss of biodiversity in Brazil. The country's still increasing demand for food and biofuels leads to the continued expansion of agricultural land [21,24]. Some researchers claimed that Brazil does not have enough potential for food production increase and restoration of deforested areas [25], while others argued that Brazil has enough land to meet increased demand for agricultural products, at the same time saving enough land for nature conservation [26]. This makes Brazil a relevant case study for LUC modeling.

## 2. Materials and Methods

### 2.1. Workflow Summary

Our workflow consists of three main steps (Figure 1). For coupling and comparison we harmonized the LU classes and input datasets of GLOBIOM-Brazil and PLUC (Table 1) and used demand for LU classes projected by GLOBIOM-Brazil as an input for PLUC, aggregated by (i.e., summed over) macro regions (Figure 1). After harmonization and demand aggregation, we ran the coupled model for the period from 2007 to 2030. The results of the coupled model are upscaled (i.e., resampled) to a resolution of 0.5 by 0.5 decimal degrees (the resolution of GLOBIOM-Brazil) and compared to the results of GLOBIOM-Brazil. To assess the accuracy of the projections made by the coupled model and GLOBIOM-Brazil we validated the results using a time series of land-use maps up to 2015 of the state of Mato Grosso [21,27] as observational data. For this validation, we harmonized the LU classes again, now between GLOBIOM-Brazil, the coupled model and these observational data (Table 1). Finally, we used spatially aggregated validation metrics [28] and a spatially explicit comparison of both LUC projections with the observational data to quantify and describe the effect of model coupling. Details of the models and these three workflow steps are provided in the next subsections.

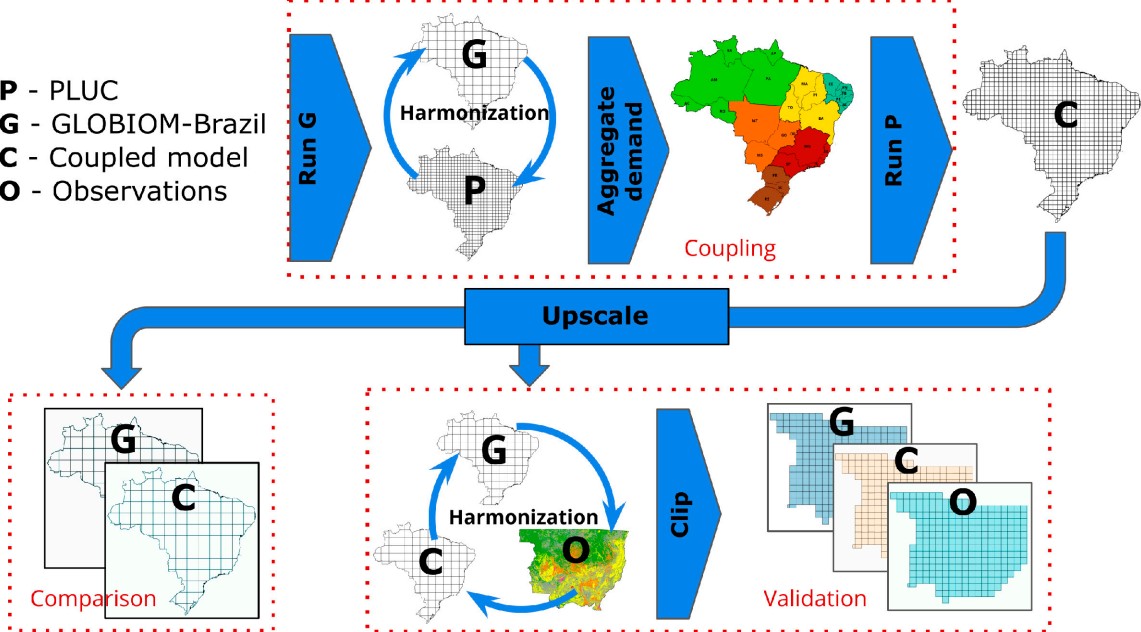

**Figure 1.** Workflow consisting of three main steps (dotted boxes): coupling, comparison, and validation. Herein, aggregation means summing over all grid cells in a region, whereas upscaling means resampling a raster to a coarser resolution.

**Table 1.** Harmonization for model coupling between land-use classes in GLOBIOM-Brazil and PLUC. For the final coupled-model classes, it is indicated whether or not they are used as demand-driven land-use (LU) classes in PLUC.

| GLOBIOM-Brazil | PLUC | Coupled Model | Observational Data | Validation |
|---|---|---|---|---|
| Cropland* | Cropland | Cropland[Δ] | Cropland*** | Cropland |
|  | Sugarcane | Sugarcane[Δ] | Sugarcane | Sugarcane |
| Grassland | Rangeland | Pasture[**Δ] | Pasture | Pasture |
|  | Planted pasture |  |  |  |
| Planted forest | Planted forest | Planted forest[Δ] | Forest | Forest |
| Managed forest | Natural forest | Natural forest |  |  |
| Forest regrowth |  |  |  |  |
| Mature forest |  |  |  |  |
| Natural land | Grass and shrubs | Natural non-forest land | Cerrado | Natural non-forest land |

* Cropland in GLOBIOM-Brazil is subdivided into barley, dry beans, cassava, chick peas, corn, cotton, groundnut, millet, potatoes, rapeseed, rice, soybeans, sorghum, sugarcane, sunflower, sweet potatoes, wheat, and oil palm. ** Internally, the coupled model distinguishes between rangeland and planted pasture, but in the model output. they are merged into a single LU class (pasture). *** Cropland in the observational data is divided into the cropping combinations cotton-fallow, soy-corn, soy-cotton, soy-fallow, soy-millet, and soy-sunflower. Δ Demand-driven in coupled model.

## 2.2. The LUC Models GLOBIOM-Brazil and PLUC

GLOBIOM-Brazil is a regional adaptation of the bottom-up partial equilibrium (PE) model GLOBIOM [29] (Table 2). It has been used in previous studies for assessing the influence of government policies (e.g., the Soy moratorium [30] and the Forest code [17]) on LUC in Brazil, and has served to support Brazil's Nationally Determined Contributions to the Paris Agreement in Global Change. GLOBIOM-Brazil projects LUC with a time step of 10 years (Figure 2), meaning that the equilibrium is solved sequentially for periods of 10 years, each time updating the initial situation with the equilibrium from the previous period. This is done for the whole world, across two spatial scale levels. At the first level, the world is divided into 30 regions, one of which is Brazil. At the second level, all regions are divided into a grid of 2 by 2 degrees (~220 km by 220 km at the equator). Brazil, however, is modelled with more detail, using a grid of 0.5 by 0.5 decimal degrees (~ 55 km × 55 km at the equator), resulting in 3001 cells for the whole country.

In these grid cells, crop, livestock and wood production are explicitly represented (Table 1). In total, 18 crop types are represented in the model, together capturing 86% of the cultivated area in Brazil in the year 2000 [17]. The production quantities, land use areas of the crops, livestock and wood are modelled at the grid cell level, whereas final demand, processing quantities, prices, and trade are computed at the regional level. The drivers of quantity of change (regional level) are population and GDP growth, diets, bioenergy demands, price elasticities, processing costs, and trade costs. The drivers of the location of change (local level) are land use productivity (potential yield), production costs, and transportation costs. For the transportation costs, the destination is the population agglomeration in the Southeast of Brazil for primarily domestically consumed products, and the nearest seaport for export products. At the local level, LUC is subject to constraints posed by protected areas and indigenous reserves. The processes implemented at the two scale levels are tightly coupled such that regional factors determine the allocation of land use in the grid cells and, at the same time, the local constraints determine the outcomes at the regional level.

PLUC is a spatially-explicit demand-driven LUC model based on the cellular automata principle, that has been applied to several case studies in multiple countries [14,20,31,32] (Table 2). When we refer to PLUC in the rest of this paper, we mean the version for Brazil [14,32]. PLUC projects LUC with

a time step of one year, at a single scale level, a grid with grid cells of 5 km by 5 km [14,32], resulting in 342,815 cells for the whole country, i.e., about 100 times more grid cells than GLOBIOM-Brazil.

　　PLUC has five active land-use classes[2] (Table 1): cropland, sugarcane, planted pasture, rangeland, and planted forest. Two LU classes are static: urban and water. Finally, four passive LU classes exist: natural forest, grass and shrubs, abandoned agricultural land and bare soil, of which the latter covers only a small area. Demands for the active LU classes are model inputs at the level of six macro regions (see [14], Figure 1 and Table 2). PLUC's state transition function solves the demands for all active LU classes in space at each time step for each of these six regions independently. If the demand for an active LU class for a macro region is higher than its current supply, cells are allocated to this LU class, starting from cells with the highest suitability value in the macro region, until demand for this LU class is fulfilled. This expansion is subject to constrains posed by static LU classes, already allocated active LU classes, protected areas, and indigenous reserves. If the demand is lower than the current supply, cells are removed from this LU class, starting from cells with the lowest suitability value in the macro region, until demand for this LU class is fulfilled. As a consequence, passive LU classes may expand or contract passively, driven by the changes in active LU classes (see [14,20] for more details). PLUC allocates the LU classes in a specific order: planted pasture, planted forest, sugarcane, rangeland, and cropland [14]. This order is important because previously allocated LU classes are excluded from allocation to keep their demands and supplies matching.

**Figure 2.** Temporal extent and resolution of GLOBIOM-Brazil, PLUC, and the coupled model. Each black dot surrounded by a circle is an initial land use map (model input); each single black dot represents a model output. The black box indicates the validation period.

**Table 2.** Overview of the methodological differences between the two models that are coupled to each other: GLOBIOM-Brazil and PLUC.

| | GLOBIOM-Brazil | PLUC |
|---|---|---|
| **Modeling paradigm** | partial equilibrium (PE) model | demand-driven model |
| **Demand for land in Brazil** | endogenous | exogenous per macro region |
| **Allocation of land classes** | optimization with linear programming | fixed order, calibrated |
| **Forest protection in land allocation** | illegal deforestation possible, Forest Code enforced in Amazon only | protected areas and indigenous reserves excluded from expansion |
| **Spatial resolution of finest scale** | 0.5 by 0.5 decimal degrees | 5 kilometer by 5 kilometer |
| **Temporal resolution** | 10 years | 1 year |
| **Data type of output maps** | scalar | nominal |
| **Land-use classes in output maps** | see Table 1 | |

　　The total suitability map for each active LU class in PLUC is a weighted sum of the maps of suitability factors for this class. Suitability factors used for Brazil are: number of neighboring grid cells with the same LU class, travel time to hubs, potential yield, length of growing season (indicating the potential for double cropping), and conversion elasticity. In the suitability factor 'travel time to hubs', the characterization of a hub depends on the LU class, e.g., hubs are slaughterhouses for pasture but silos for cropland. The weights of the suitability factors and the order of allocation of the active LU classes have been calibrated based on historic non-spatial data of LU trends per state ([14], p. 570).

---

[2]　The classification is in fact a mix of land use and land cover classes, but for simplicity we refer to them as land use classes throughout this paper.

### 2.3. Model Coupling

Harmonization of model components is necessary for model coupling, comparison and validation [10]. Firstly, LU classes used in the models should be harmonized to allow comparison of outputs and the usage of one model's outputs as the other model's inputs. Secondly, input datasets and scenarios should be made consistent across models. Finally, the level of exchange of variables (output of model one and input of model two) should be matched; in our case, for the land demand.

We applied all three types of harmonization. Table 1 shows the harmonization of LU classes between PLUC and GLOBIOM-Brazil for comparison and coupling. This results in the following LU classes in the coupled model: cropland, sugarcane, pasture, planted forest, natural forest, and natural non-forest land. For the remaining classes (see Section 2.2) the models were not coupled.

To harmonize input datasets, the map of protected areas and indigenous reserves in PLUC was replaced by the map used in GLOBIOM-Brazil. Replacing PLUC's initial land-use with the map describing initial state of GLOBIOM-Brazil model was not possible because GLOBIOM-Brazil's initial land-use map exists only at the resolution of GLOBIOM-Brazil, which makes its application in PLUC impossible. Other model comparison studies also do not apply a common initial map (e.g., [1,12]), potentially having a large influence on the model results, making it interesting to include this effect in the model comparison (see Section 2.5). As the common scenario for both models, we took the "middle of the road" Shared Socioeconomic Pathways (SSP2) [33]. We used existing LUC projections from GLOBIOM-Brazil with this scenario, from 2000 to 2050 [17]. These projections and the input data used to obtain them are available via Pangea [34].

For coupling PLUC with GLOBIOM-Brazil, we used the demand for the harmonized LU classes projected by GLOBIOM-Brazil as an input for PLUC. Hereto, the demand projected by GLOBIOM-Brazil for the four active LU classes in PLUC was aggregated from the 0.5-by-0.5-decimal-degree grid cells to the six macro regions (Figure 1). The algorithm of demand aggregation is as follows: We summed the areas of change in the active land-use classes over the macro regions for the pairs of output years of GLOBIOM-Brazil, 2000–2010, 2010–2020, and 2020–2030. Next, to harmonize the temporal resolutions of GLOBIOM-Brazil and PLUC (Figure 2), we calculated the annual LUC per LU class by linear interpolation between these years. Finally, we calculated the demands for PLUC in total area per LU class by adding this annual LUC to the land use area of the corresponding LU class in PLUC's initial map of 2006. To divide GLOBIOM-Brazil's grassland class into pasture and rangeland (Table 1), we used the relative occurrences of these classes projected by Verstegen et al. [14].

In summary, in the outputs of the coupled model, GLOBIOM-Brazil determines the total amount of change of each active land-use class for the whole of Brazil, GLOBIOM-Brazil determines in which macro region these changes are allocated, PLUC determines in which grid cell within the macro region the changes in the active land-use classes are allocated, and, thereby, PLUC determines the location as well as the total amount of change in each passive land-use class.

### 2.4. Model Comparison

We compared the LUC projections of GLOBIOM-Brazil and the coupled model to evaluate the effect of model coupling. Comparing the coupled model with PLUC would not be meaningful because PLUC requires demands as an input. When demands are provided that are different from GLOBIOM-Brazil's, the LUC projections of PLUC will be different from the ones by GLOBIOM-Brazil by definition and when the same demands are taken, PLUC is not independent from GLOBIOM-Brazil anymore, leading to a biased comparison.

For the comparison, we ran the coupled model for the period from 2007 to 2030 (Figure 2). The output of the coupled model is, correspondingly to PLUC's outputs, a time series of grids with a cell size of 5 km by 5 km, in which each cell has one LU class, i.e., a nominal map (Figure 3). The outputs of the GLOBIOM-Brazil model are grids with a resolution of 0.5 by 0.5 decimal degrees, in which each grid cell contains values representing the area of each LU class, i.e., a set of scalar maps (Figure 3). To allow comparison, we upscaled the outputs of the coupled model to the resolution of the

output of GLOBIOM-Brazil, thereby calculating the fraction of all LU classes in each 0.5 by 0.5 decimal degrees cell (Figure 3). We compared the model outputs at this resolution. For the validation (see the next section), this was done for the period from 2007 to 2015 and for the projections (results in Section 3.3), this was done for the period from 2010 to 2030 (Figure 2).

The initial system state of a LUC model influences the LUC projections, so we evaluated differences between the initial system states of the two models. The initial system states of the two models are not directly comparable because the initial map of GLOBIOM-Brazil is a set of scalar maps for the year 2000 whereas the initial map of PLUC is a nominal map for the year 2006 (Figure 2). Therefore, we computed the total area of each harmonized LU class per macro region for 2000 for GLOBIOM-Brazil and for 2006 for PLUC (initial state of the coupled model) as well as the total area for GLOBIOM-Brazil for 2006 based on interpolation between the initial map of GLOBIOM-Brazil and its projection for 2010.

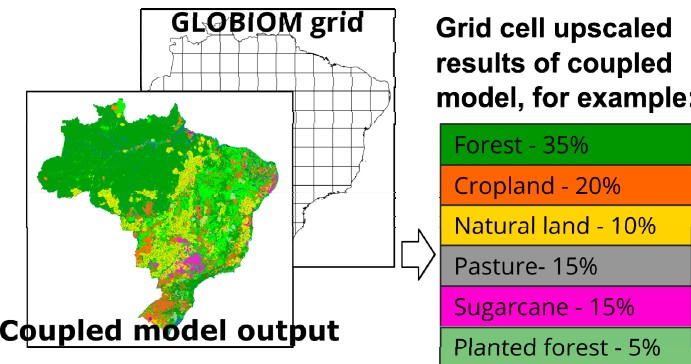

**Figure 3.** Upscaling of the results of PLUC in the coupled model to GLOBIOM-Brazil resolution, implying a change from nominal (categorical) to scalar (fractional) data as shown on the right-hand side. Note that the image of the output map contains the non-harmonized classes; therefore, the colors of the grid cells do not correspond to the LU class colors used throughout this paper.

## 2.5. Model Validation

In the validation, we evaluated the performance of GLOBIOM-Brazil and the coupled model. Hereto, we compared the LUC projected by the two models from 2007 to 2015 with observations of LUC derived from a time series of LU maps developed by the Brazilian Space Agency (INPE) [21][3]. These time series contain data from 2001 to 2016 and are available via Pangea [27]. The validation was performed per LU class. Hereto, we performed harmonization between GLOBIOM-Brazil and the coupled model on the one side and the time series of land-use maps we used as observational data on the other side (Table 1). This results in a reduction from six to five harmonized LU classes.

The observations [21] are for the state of Mato Grosso, located in the west of Brazil. Mato Grosso is around 900,000 square kilometers, i.e., almost three times the size of Germany, and the third-largest state of Brazil (about 10% of the total area of Brazil). The so-called arc of deforestation of the Amazon, where agriculture is expanding rapidly [21], runs through Mato Grosso. The size and dynamic nature of the state make that we consider it suitable as a validation area. Because the observations are for the state of Mato Grosso, the model results are also clipped to Mato Grosso for the validation.

The performance indicators typically used for the validation LUC models, such as Kappa simulation [35], are not applicable in our case, because due to the upscaling, we have five scalar variables (area of cropland, sugarcane, pasture, forest and natural non-forest land) instead of one nominal variable per grid cell. For that reason, we used the Root Mean Square Error (RMSE),

---

3  LU derived from remote sensing is not directly observed. Reflectance is observed by the sensor and this reflectance is translated into land use. Therefore, technically, the time series of LU maps are a modelled variable too. However, in the context of our work, the time series of LU maps represent the independent land use situation to which we compared our model results; therefore, we refer to them as 'observations'.

the Nash–Sutcliffe model efficiency coefficient (NSE), and the composite operator for the model validation, separately for each of the five LU classes.

The RMSE is a frequently used metric to express the differences between a modelled variable and an observed variable (Equation (1)):

$$RMSE_i \; = \; \sqrt{\frac{\sum_{n=1}^{N}(\hat{z}_{n,i} - z_{n,i})^2}{N}} \tag{1}$$

where $n$ is the grid cell with $n = 1, 2, \ldots N$, $i$ is the LU class with $i = 1, 2, \ldots, I$. Furthermore, $\hat{z}_{n,i}$ is the modelled system state for LU class $i$, here the modelled area of LU class $i$ in grid cell $n$, $z_{n,i}$ is the observed system state for LU class $i$, and $N$ is the total number of observations. In our case, the total number of 0.5 by 0.5 decimal degree grid cells in Mato Grosso was 248. a difference in RMSE between GLOBIOM-Brazil and the coupled model may be caused by either the difference in model structure (single vs. coupled) or the difference in the initial land use map (see Section 2.3). The two sources cannot be completely disentangled but to get an impression of the contribution of the two components, we computed the RMSE for the *total areas* of the LU classes in 2015 (for a large part determined by the initial map) as well as for the projected *change* in the areas of LU classes between 2007 and 2015 (mainly determined by the model structure) of all land-use classes. Thus, in total, we computed five (LU classes) times two (models) times two (system state variables), giving twenty RMSE values.

The RMSE is scale-dependent, so it has a relative meaning only. Therefore, we merely used it to compare the performance between GLOBIOM-Brazil and the coupled model. To also get an idea of the absolute performance of both models, we used the NSE [28] (Equation (2)), a metric that is widely used in the hydrological modeling domain, but in principle, is applicable to any kind of model.

$$NSE_i \; = \; 1 - \frac{\sum_{n=1}^{N}(\hat{z}_{n,i} - z_{n,i})^2}{\sum_{n=1}^{N}(z_{n,i} - \overline{z_i})^2} \tag{2}$$

where, in addition to the variables used in Equation (1), $\overline{z_i}$ is the mean over all observations for LU class $i$. The NSE ranges from $-\infty$ to 1. An NSE of 1 indicates a perfect match between modeled and observed system state. At an NSE of 0, the model projections are as accurate as the mean of the observed data. An NSE of less than 0 occurs when the residual variance (the numerator) is larger than the data variance (the denominator), implying that the observed mean is a better predictor than the model.

In the RMSE and the NSE, errors in quantity and errors in location are mixed together in a single metric. To disentangle these two types of errors, we also apply the composite operator [36]. The composite operator computes, per LU class, the total agreement between modelled and observed fraction in a cell, $a_{n,i}$ (Equation (3)). The disagreement (modelled fraction minus agreement) is then divided among the other LU classes, proportionally to the disagreement of each of these other LU classes, to obtain the disagreement per LU class pair, $d_{n,i,j}$ (Equation (4)).

$$a_{n,i} \; = \; MIN(\hat{z}_{n,i}, z_{n,i}) \tag{3}$$

$$d_{n,i,j} \; = \; (\hat{z}_{n,i} - a_{n,i}) \cdot \left[ \frac{z_{n,j} - a_{n,j}}{\sum_{j=1}^{J} z_{n,j} - a_{n,j}} \right], \; for \; i \neq j \tag{4}$$

where $j$ is another LU type than $i$, but out of the same set, $j = 1, 2, \ldots, I$.

A matrix with the total quantity (dis)agreement and total location (dis)agreement is obtained by averaging the agreement and all disagreements of LU class $i$ over all grid cells $N$. In Equations (3) and 4, $\hat{z}_{n,i}$ and $z_{n,i}$ have to meet the condition $\sum_{i=1}^{I} \hat{z}_{n,i} = 1$, respectively $\sum_{i=1}^{I} z_{n,i} = 1$ [36]. Therefore, we needed to edit our system state variables in two ways: 1) we converted from area per LU class to fraction of LU class per grid cell by dividing by the cell size, and 2) we added a LU class 'other' to the total set of LU classes. The latter is necessary because we did not harmonize all LU classes between

the two models, so without 'other', the total of the LU class fractions would not be one. The new LU class other contains, for example, urban areas, water, and abandoned agricultural land. The class other was assigned the remaining cell fraction, i.e., one minus the fractions of all harmonized LU classes. Since this class is not harmonized across the two models, a low agreement in quantity and location is expected for it.

In addition to the RMSE, the NSE and the composite operator, which are a spatially aggregated metrics, we calculated spatially explicit differences in LUC between the time series of land-use maps of Brazil and the outputs of the two models, i.e., the difference in projected change per 0.5 by 0.5 decimal degree cell. This allows us to see where in space the projection accuracy is high or low.

## 3. Results

### 3.1. Initial Land Use Maps

In the initial state of the coupled model, i.e., the year 2006, there is 460 Mha of natural forest, 159 Mha of natural non-forest land, 159 Mha of pasture, 42.1 Mha of cropland, 12.8 Mha of planted forest and 5.91 Mha of sugarcane (Figure 4). The interpolated areas of GLOBIOM-Brazil for the same year are 16 Mha (3.5%) lower for forest, 97 Mha (61%) lower for natural non-forest land, 71 Mha (44%) higher for pasture, 3.2 Mha (7.7%) higher for cropland, 5.8 Mha (46%) lower for planted forest, and 0.67 Mha (11%) higher for sugarcane.

The large difference in natural non-forest land has its main origin in the Northeast Cerrado, Northeast Coast, and Southeast (Figure 4). In these three macro regions, GLOBIOM-Brazil has much lower areas of natural non-forest land than the coupled model, whereas the coupled model has much lower areas of natural forest. The difference in pasture area mainly stems from the Northeast Cerrado, Center West Cerrado, and Southeast. GLOBIOM-Brazil always gives a larger pasture area than the coupled model.

The areas of forest also show large differences in several macro regions, but, in contrast to the differences in pasture and natural non-forest land, it varies which of the models suggests the largest area. Therefore, these differences per macro region level out at the scale of the whole of Brazil.

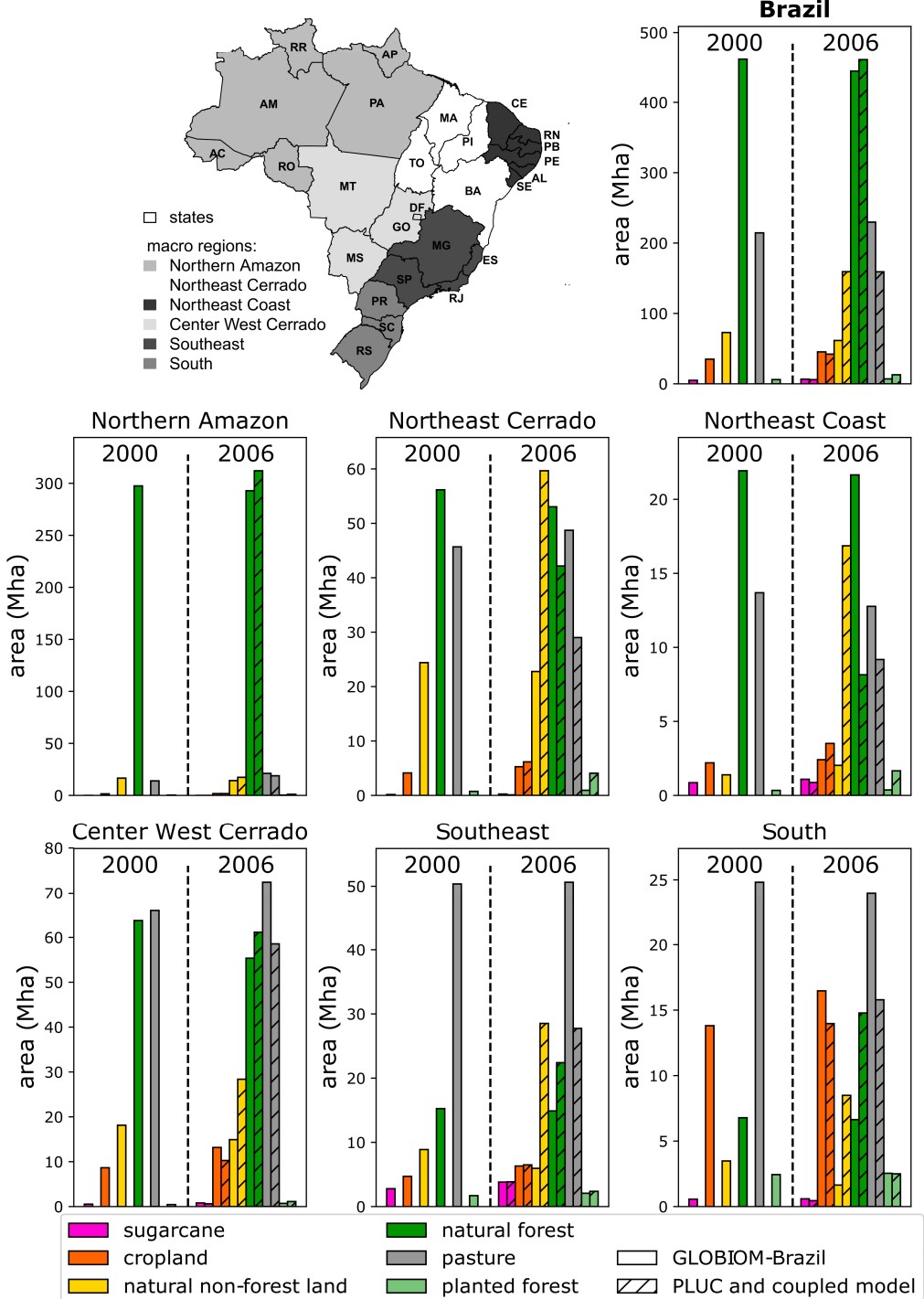

**Figure 4.** Total area of each harmonized land-use class in the initial map of GLOBIOM-Brazil (2000) and in the initial map of the coupled model (2006) for each of the six macro regions and for the whole of Brazil. For comparison, the total area of each harmonized land-use class projected by GLOBIOM-Brazil for 2006 is added. Abbreviations of the states in the map are: AC = Acre, AL = Alagoas, AM = Amazonas, AP = Amapá, BA = Bahia, CE = Ceará, DF = Distrito Federal, ES = Espírito Santo, GO = Goiás, MA = Maranhão, MG = Minas Gerais, MS = Mato Grosso do Sul, MT = Mato Grosso, PA = Pará, PB = Paraíba, PI = Piauí, PR = Paraná, RJ = Rio de Janeiro, RN = Rio Grande do Norte, RO = Rondônia, RR = Roirama, RS = Rio Grande do Sul, SC = Santa Catarina, SE = Sergipe, SP = São Paulo, TO = Tocatins.

### 3.2. Validation of Land Use Change in Mato Grosso

The performance of GLOBIOM-Brazil and the coupled model were quantified using the composite operator, the RMSE of total LU, the RMSE of LUC, and the NSE. The composite operator shows that the most accurate LU classes in terms of quantity (area) are pasture projected by the coupled model and cropland projected by GLOBIOM-Brazil (Tables 3 and 4, compare totals). Pasture area is highly overestimated by GLOBIOM-Brazil (55% higher than in the observations) while cropland area is underestimated by the coupled model (−17%). Both models overestimate the total area of sugarcane and other land (the category that contains all LU other than the five harmonized classes from Table 1), but they underestimate the total area of natural non-forest land. Forest area is overestimated by the coupled model (+33%) but underestimated by GLOBIOM-Brazil (−15%).

**Table 3.** Comparison of the modelled LU by the GLOBIOM-Brazil and the observed LU using the composite operator [36].

| | | **observations** | | | | | | |
|---|---|---|---|---|---|---|---|---|
| | | sugarcane | cropland | natural non-forest land | forest | pasture | other | total |
| **GLOBIOM-Brazil** | sugarcane | **0.20%** | 0.09% | 0.45% | 0.10% | 0.00% | 0.00% | 0.84% |
| | cropland | 0.07% | **7.55%** | 2.27% | 1.13% | 0.33% | 0.03% | 11.38% |
| | natural non-forest land | 0.00% | 0.64% | **6.51%** | 0.67% | 0.10% | 0.02% | 7.95% |
| | forest | 0.01% | 0.66% | 4.35% | **22.87%** | 0.68% | 0.02% | 28.59% |
| | pasture | 0.03% | 1.74% | 8.07% | 6.28% | **25.43%** | 0.03% | 41.58% |
| | other | 0.02% | 0.78% | 4.60% | 2.78% | 0.32% | **1.17%** | 9.66% |
| | total | 0.33% | 11.46% | 26.24% | 33.83% | 26.87% | 1.27% | 100.00% |

**Table 4.** Comparison of the modelled LU by the coupled model and the observed LU using the composite operator [36].

| | | **observations** | | | | | | |
|---|---|---|---|---|---|---|---|---|
| | | sugarcane | cropland | natural non-forest land | forest | pasture | other | total |
| **coupled model** | Sugarcane | **0.25%** | 0.11% | 0.12% | 0.01% | 0.07% | 0.00% | 0.56% |
| | Cropland | 0.02% | **7.00%** | 1.21% | 0.42% | 0.84% | 0.03% | 9.52% |
| | natural non-forest land | 0.00% | 0.47% | **12.90%** | 0.10% | 0.58% | 0.04% | 14.10% |
| | Forest | 0.02% | 2.38% | 6.18% | **32.14%** | 4.29% | 0.10% | 45.12% |
| | Pasture | 0.03% | 0.81% | 3.59% | 0.68% | **19.74%** | 0.07% | 24.91% |
| | Other | 0.01% | 0.68% | 2.24% | 0.49% | 1.35% | **1.03%** | 5.80% |
| | Total | 0.33% | 11.46% | 26.24% | 33.83% | 26.87% | 1.27% | 100.00% |

In terms of location, the highest producer's accuracy (95%) is obtained for forest projected by the coupled model and pasture projected by GLOBIOM-Brazil (Tables 3 and 4, divide cell values in the diagonal by the totals of the observations). This may seem in contradiction with the high

overestimation of forest quantity by the coupled model and pasture quantity by GLOBIOM-Brazil, but it is not: consider that when the whole map is projected to be pasture, at least all locations of pasture in the observed map are correctly projected (producer's accuracy of 100%, but low user's accuracy). The lowest producer's accuracy (25%) is obtained for natural non-forest land projected by GLOBIOM-Brazil; this class is misallocated mainly to pasture. The highest user's accuracy (91%) is obtained for natural non-forest land by the coupled model (Tables 3 and 4, divide cell values in the diagonal by the totals of the model), whereas the lowest user's accuracies are obtained for the class other (GLOBIOM-Brazil 12% and coupled model 18%) and for sugarcane by GLOBIOM-Brazil (25%).

The combined effect of quantity (dis)agreement and location (dis)agreement is reflected in the RMSE. For the LUC between 2007 and 2015, the RMSE of the coupled model projections is lower than RMSE of the GLOBIOM-Brazil projections for all LU classes (Figure 5). The relative improvement of the RMSE of the coupled model projections with respect to the RMSE of GLOBIOM-Brazil projections is 80% for natural forest, 69% for pasture, 55% for sugarcane, 32% for natural non-forest land, and 31% for cropland.

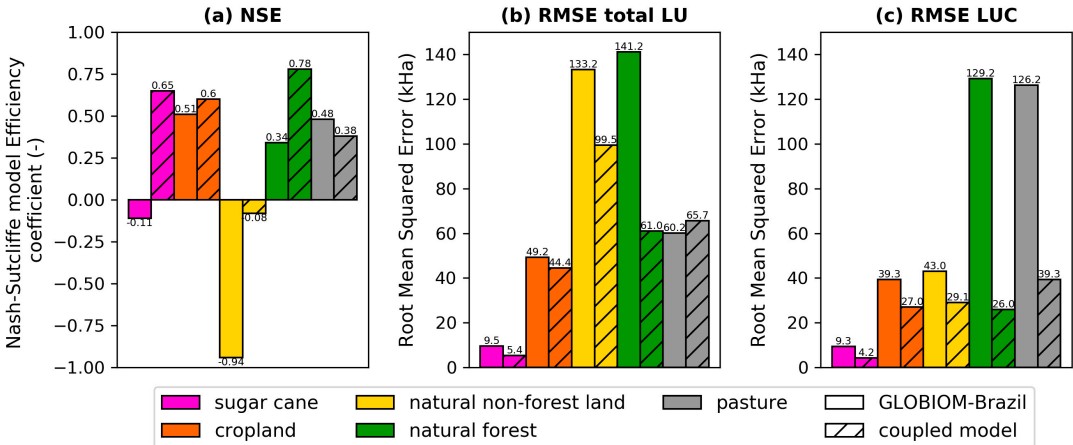

**Figure 5.** (**a**) The Nash–Sutcliffe efficiency (NSE), (**b**) Root Mean Squared Error (RMSE) of total land use in 2015 and (**c**) of land use change from 2007 to 2015. The three validation metrics are computed for Mato Grosso between the model projections (with GLOBIOM-Brazil and the coupled model) and the observations.

For natural forest and pasture, the differences in performance are the largest and occur all over Mato Grosso (Figure 6). For cropland and sugarcane, differences are more concentrated mainly in the center of Mato Grosso (Figure 6). For natural non-forest land, the differences between GLOBIOM-Brazil and the observations occur throughout Mato Grosso, whereas the differences between the coupled model and the observations are found in the southern half of the state (Figure 6).

For the total LU in 2015, the RMSE of the coupled model projections is lower than the RMSE of the GLOBIOM-Brazil projections for all LU classes except pasture (Figure 5). The relative improvement of the RMSE of the coupled model projections with respect to the RMSE of GLOBIOM-Brazil projections is 57% for natural forest, 43% for sugarcane, 25% for natural non-forest land, and 10% for cropland. For pasture, the RMSE of the total land use in 2015 is 9% higher (worse) for the coupled model compared to GLOBIOM-Brazil.

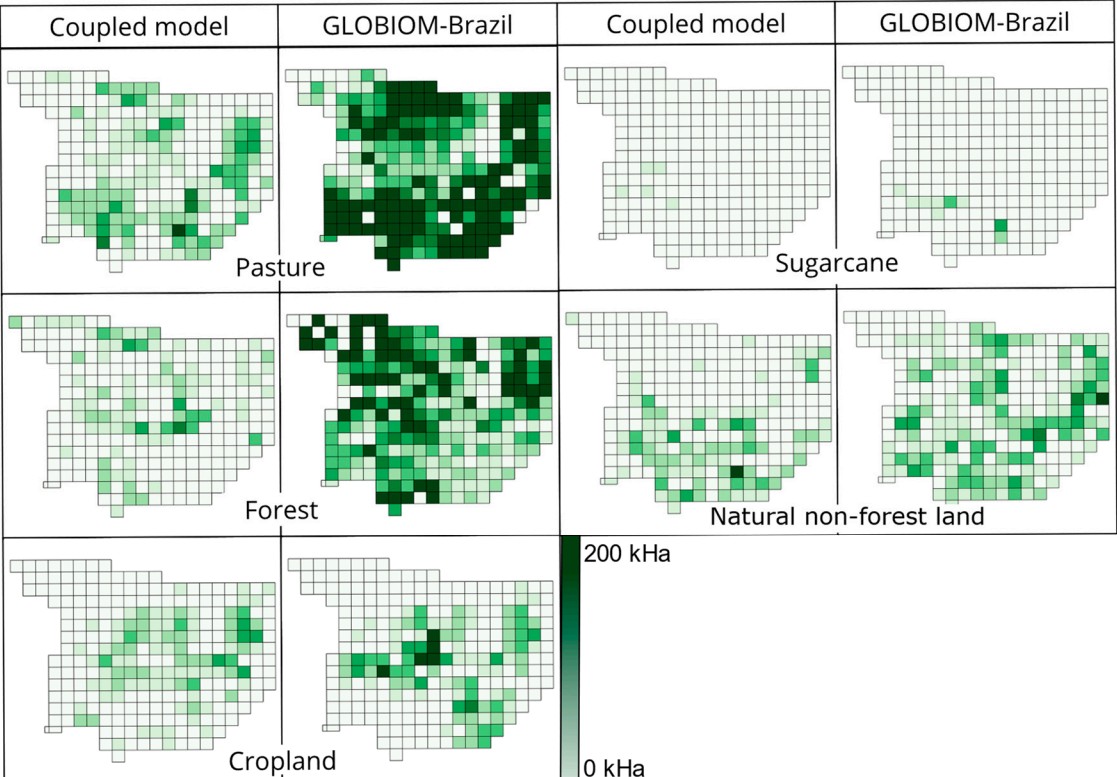

**Figure 6.** Differences between LUC projected by the models and LUC in the observational data [21,27] for Mato Grosso for 2007–2015. The darker colors indicate larger differences, i.e., higher model errors.

The NSE metric, which is corrected for the total variance within the projections of one LU class, gives results along the same line: the coupled model performs better for all LU classes except pasture. The increase in the NSE of the coupled model projections with respect to the NSE of GLOBIOM-Brazil projections is, 86 percent point for natural non-forest land, 76 percent-point for sugarcane, 44 percent point for natural forest and 9 percent point for cropland. For pasture, the NSE of the total LU in 2015 is 10 percent point lower (worse) for the coupled model compared to GLOBIOM-Brazil.

These results indicate that the combined effect of the initial land use map and the model structure of the coupled model results in better projections for sugarcane, cropland, forest, and natural non-forest land. For natural non-forest land, however, these improved projections are still not trustworthy (NSE below zero). For pasture, GLOBIOM-Brazil generates the most accurate projections.

### 3.3. Comparison of Land-Use Change Projections up to 2030

GLOBIOM-Brazil and the coupled model project LUC patterns up to 2030 that differ in various respects (Figure 7). In general, the LUC projected by the coupled model is concentrated, whereas the changes projected by GLOBIOM-Brazil are widespread. Below, we discuss the most important differences between the results of the two models per LU class, in the order in which the LU classes are shown in Figure 7. We use the term 'small difference' for 1-50 kHa and 'large difference' for > 50 kHa, corresponding to the distinction in Figure 7 made by the black dots.

The coupled model shows hotspots (a hotspot is a contiguous group of grid cells with large changes) of pasture land contraction in the states of Paraná and São Paulo, while hotspots of expansion are projected in Rondônia, Roraima and the eastern part of Pará. GLOBIOM-Brazil projects the highest pasture land contraction in the states of Mato Grosso do Sul and Paraná, while the areas of significant expansion are the states of Maranhão, Mato Grosso and Pará. The direction of change is typically the same (i.e., both models project expansion or both project contraction). The differences in pasture area change between the two models are zero in 17%, small in 62%, and large in 21% of all grid cells.

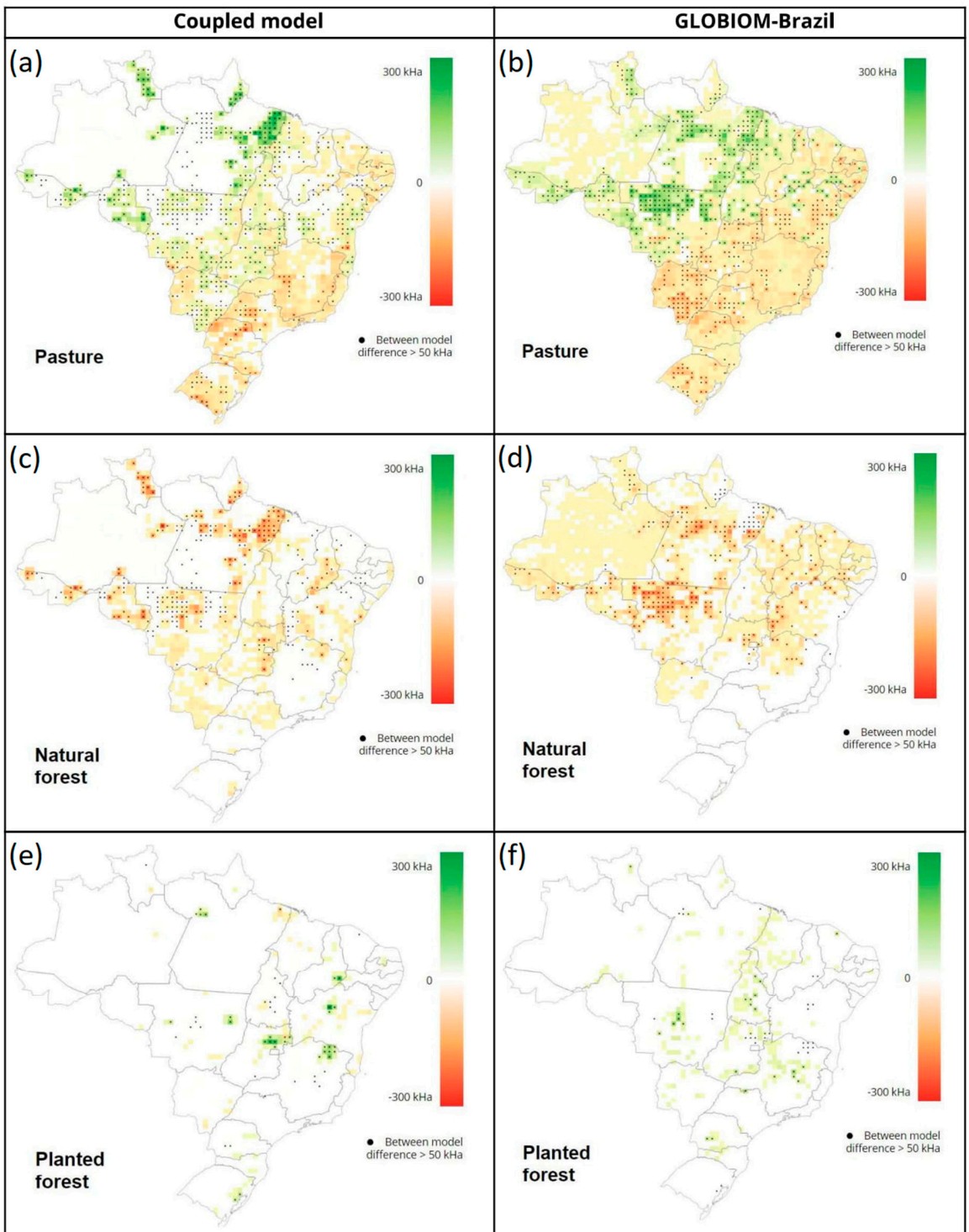

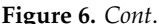

**Figure 7.** Projected land-use changes for the period 2010 to 2030 by the coupled model (left) and GLOBIOM-Brazil (right). Projections are compared per land-use class: pasture (**a,b**), natural forest (**c,d**), planted forest (**e,f**), cropland (**g,h**), sugarcane (**i,j**), and natural non-forest land (**k,l**). Colors indicate the amount of increase (green) or decrease (red) per grid cell over the time periods. The black dots indicate grid cells with large (> 50 kHa) differences in projected changes between the two models. The threshold of 50 kHa was based on the frequency distribution of the data.

For natural forest, the two models agree on the locations of hotspots of deforestation in Mato Grosso, Pará and Rondônia. In addition, GLOBIOM-Brazil projects deforestation hotspots in Bahia, and the coupled model in other parts of the Amazon (mainly Amapá and Roirama). In these parts of the Amazon, GLOBIOM-Brazil projects low percentages of deforestation across all grid cells instead of clear hotspots. In addition, GLOBIOM-Brazil projects small reforestation areas. There is no difference in forest area change between the two models in 37%, a small difference in 53%, and a large difference in 10% of the grid cells.

The projected locations of expansion of planted forest differ considerably between the models. The coupled model allocates planted forest mostly in the states of Bahia, Goiás, Minas Gerais and Piauí. GLOBIOM-Brazil shows that planted forest increases in the states of Amazonas, Goiás, Maranhão, Mato Grosso do Sul, Minas Gerais, Pará, Paraná and Tocantins. Because of the small area of this LU class, there is (despite the differences in locations of change) no difference in planted area change between the two models in 88%, a small difference in 10%, and a large difference in 2% of the grid cells.

The coupled model projects cropland contraction at several locations in Goiás, Mato Grosso, Mato Grosso do Sul, Pará and São Paulo. Furthermore, it projects cropland expansion at locations in all states except in the Northern Amazonas region. Major expansion can be observed in states of Piauí, Mato Grosso, Minas Gerais and Rio Grande do Sul. In GLOBIOM-Brazil, the locations of cropland contraction are spread over a larger area than in the coupled model, mainly in Amazonas, Bahia, Mato Grosso, Tocantins and the North-East coast region. Thus, both the amount of change per state and the allocation patterns of cropland vary between the models. Sometimes, even the direction of changes differs, e.g., the coupled model projects expansion in the center of Mato Grosso, whereas GLOBIOM-Brazil projects contraction, and the reverse is true for the South of Mato Grosso do Sul. There is no difference in cropland area change between the two models in only 7%, a small difference in 83%, and a large difference in 10% of the grid cells.

Sugarcane production areas are located only in certain states. In the coupled model, change in sugarcane area is observed in only 10% of Brazil. By the coupled model, sugarcane cultivation areas are projected to be abandoned almost nowhere while expansion is projected mainly in Goiás, Mato Grosso do Sul, Paraná, and São Paulo. GLOBIOM-Brazil projects changes in 40% of the country, with contraction and expansion in almost all the states. The highest differences between the projections of the two models occur in the states of Goiás, Mato Grosso, Mato Grosso do Sul, Minas Gerais, Paraná, São Paulo and in the North-East coast region. There is no difference in sugarcane area change between the two models in 61%, a small difference in 35%, and a large difference in 4% of the grid cells.

The coupled model projections show natural non-forest land decrease in all states. The hotspots of shrinkage are in Goiás, Mato Grosso, São Paulo and Piauí. Unlike the coupled model, GLOBIOM-Brazil projects both positive and negative change in natural non-forest land in all the states. The major differences between the coupled model and GLOBIOM-Brazil can be observed in states of Bahia, Goiás, Maranhão, Mato Grosso, Mato Grosso do Sul, Piauí and Tocantins. There is no difference in natural non-forest land change between the two models in 24%, a small difference in 65%, and a large difference in 11% of the grid cells.

## 4. Discussion

### 4.1. Interpretation of the Results

In the initial system state, i.e., input LU in the year 2006, large differences in the total areas of natural non-forest land and of pasture were observed between GLOBIOM-Brazil and PLUC. Semantics are likely to be the cause of this difference, as explained in the following. Differences in natural non-forest land occurred mainly in the Northeast Cerrado and Northeast Coast, which are largely in the Caatinga biome. This biome has an arid climate and its vegetation consists mainly of Savana Estépica, characterized by stunted trees and shrubs. In PLUC, this vegetation is seen as shrubland and is therefore in the class grass and shrubs, becoming natural non-forest land in the coupled model

(Table 3). In GLOBIOM-Brazil, Savana Estépica falls into the forest class ([19], p. 17). The same happens in parts of the Cerrado biome. As another example of the influence of semantics, the differences in pasture land use are likely to be caused by how each model distinguishes grazed areas (to be considered pasture) from non-grazed areas (to be considered natural non-forest land or natural forest). Herein, PLUC follows the Brazilian Institute of Geography and Statistics (IBGE) census data [37], whereas GLOBIOM-Brazil follows the Institute for Applied Economic Research (Ipea) [38]. On the one hand, the large uncertainties in the pasture class are problematic, as this is the largest non-natural land use in Brazil [34]. On the other hand, the (often low-intensity) grazing in these areas has a relatively low environmental impact. From that perspective, it is more crucial to correctly model intensive LU classes, such as cropland.

The validation for the period 2007-2015 with the composite operator [36] implies that the coupled model has a higher user's accuracy than GLOBIOM-Brazil for all LU classes except forest, and higher quantity agreement for all LU classes except cropland and pasture. Judged by the RMSE, the coupled model projects LU (i.e., the final projected map) and LUC (i.e., the dynamics) more accurately than GLOBIOM-Brazil for all LU classes except for pasture. For pasture, LU was projected more accurately by GLOBIOM-Brazil while LUC was projected more accurately by the coupled model. This stresses the good fit of the calibrated parameter values of PLUC. This goodness of fit is interesting given the fact that time series of LU maps used in the current paper [21] were not available at the time of calibration. Instead, PLUC was calibrated using non-spatial patterns, namely the total LU area per state [14]. In view of these positive results, we advise other modelers to employ a non-spatial calibration approach to spatial-data-limited model studies, especially when the alternative is not doing a calibration at all.

Other research domains, primarily hydrology, have suggested NSE threshold values to indicate a model of sufficient quality, of 0.5 up to 0.65 [28]. Even if we use the lowest suggested threshold value, 0.5, the coupled model performs sufficiently only for the LU classes sugarcane (NSE of 0.65), cropland (0.60) and forest (0.78). The performance would be considered insufficient for pasture (0.38), and very poor (observed mean is a better predictor than the model) for natural non-forest land (−0.08). GLOBIOM-Brazil would be considered of sufficient quality for cropland only (NSE of 0.51). Yet, it is disputable whether the NSE threshold recommended in a natural science domain like hydrology is directly transferrable to LUC models. LUC models simulate human–environment interactions that are by nature much more erratic and unpredictable than natural processes [7]. This calls for the application of metrics like the NSE that corrects for the total variance, but it also requires testing across wider range of LUC models to evaluate what NSE values are reachable for LUC models before a decision on a threshold for sufficient quality can be made. Moreover, it is disputable whether a threshold is useful at all given that model sufficiency always depends on model purpose and LUC models serve a wide variety of purposes. Still, the NSE is an interesting metric as it is an absolute measure comparable across case studies. We encourage LU modelers to use this metric, especially for LUC models with scalar output, for which few applicable metrics exist.

In general, the areas exhibiting the highest differences between the projections of the two models for 2030 can be found in Cerrado biome, especially at the border with the Amazon, and along the arc of deforestation, where agricultural expansion was especially high in recent decades. These locations coincide with observed hotspots of disagreement between eleven global LUC models, reported by Prestele et al. [1]. This implies that other global models than GLOBIOM-Brazil have similar difficulties in projecting changes in highly dynamic regions like the arc of deforestation. Therefore, the conclusions presented here about the improvements attained by model coupling with GLOBIOM-Brazil may be generalizable to these other models and other highly dynamic case study areas. Agricultural expansion in these highly dynamic regions and the resulting deforestation is a key process. LUC models should be seeking to represent this process well, given that the aim of LUC model assessments is often to evaluate potential policies designed to avoid the negative effects of deforestation, such as carbon stock loss, albedo changes, increased increase soil evaporation, and biodiversity loss (e.g., [5,17,30,39]).

The observed differences between the projections of the coupled model and GLOBIOM-Brazil may have various reasons. First of all, the initial LU map is different between the models, with the largest differences for natural non-forest land and pasture. The initial situation has a large influence on the projected LU, because change occurs only in a small part of the region, so for the major part of the study area the initial situation is also the final situation. As noted by others [1], this influence of the initial situation is expected to decrease with the length of the modelled time frame. Furthermore, only in the coupled model, the initial LU affects the projections by the fact that PLUC is a constrained cellular automaton (CA). This means that the current system state is an input in the model rules that determine the next system state, causing a path-dependence effect, see, e.g., [40]. The CA principle is operationalized through the suitability factor number of neighboring grid cells with the same LU class, which has a relatively high weight in the total suitability map of all LU classes [14]. The effect of this suitability factor is the more clustered pattern of LUC, i.e., higher spatial autocorrelation, that can be observed in Figure 7.

Another reason for differences in model projections might be the order in which GLOBIOM-Brazil and the coupled model allocate LUC for different LU classes. Whereas PLUC allocates the LU classes in a specific order (see Section 2.2), transitions in GLOBIOM-Brazil are modelled according to a transition matrix ([19], p. 34). This entails that, for example, a direct transition from cropland to pasture is not possible in GLOBIOM-Brazil, while in the coupled model, it is.

Furthermore, the resolution of allocation has an effect though the detail that can be captured in the suitability factors. Herein, the higher resolution of the coupled model might have contributed to the better performance, as well as the diversity in spatial scales at which the processes are modelled (global, country, 0.5 by 0.5 decimal degree grid cells, macro regions, and 5 km by 5 km grid cells).

Another explanation, especially for the differences in deforestation pattern between the models, is that PLUC, and thus the coupled model, does not allow any deforestation in protected areas and indigenous reserves. GLOBIOM-Brazil, on the other hand, accounts for non-compliance with policies and thereby allows small areas of deforestation. This causes the large areas of small changes that can be observed in e.g., Figure 7b,h (see the large yellow regions in the Amazon). Finally, PLUC, and thereby, the coupled model, does not simulate restoration of natural non-forest land after the abandonment of agricultural land, while GLOBIOM-Brazil does. For example, expansion of natural non-forest land can be observed in the projections of GLOBIOM-Brazil (Figure 7l). This aspect is further discussed in Section 4.2.

The lower performance of the coupled model for pasture is in line with the previous model validation with spatially aggregated patterns by Verstegen et al. [14]. This validation showed that PLUC performs poorly in the allocation of rangeland (part of the pasture class in the coupled model). The authors attributed this to the difficulty to distinguish rangeland from natural grassland in remotely sensed data that was used for the calibration. If the calibration data are poor, model rules are identified poorly. In addition, there was a lack of data availability for the suitability factors (hubs and potential yield) of both rangeland and planted pasture. The confusion between pasture and natural non-forest land occurs again in the results of the current study (Table 3). Furthermore, the fact that the RMSE for LUC is lower for the coupled model but the RMSE for total LU is higher indicates that a large part of the lower performance of the coupled model for the pasture class is attributable to the initial system state. In the system state of 2006, the pasture area in the coupled model was 44% lower than in GLOBIOM-Brazil (Figure 4). We conclude from this that modelers should carefully consider if it is really necessary to include a certain class distinction (such as here rangeland versus natural grassland) when calibration data availability and input data availability are low.

GLOBIOM-Brazil has similar problems related to pasture. For the creation of its initial LU map, for the Amazon biome remote sensing data was used, whereas for the Cerrado, Caatinga, Mata Atlantica and Pantanal biomes the IBGE vegetation map was used, which has a scale of 1:5,000.000 [19]. Mato Grosso is covered partly by the Amazon biome and partly by the Cerrado biome. As grassland is the native vegetation type of the Cerrado, the coarse scale of the data to produce the initial LU map for the

Cerrado has a large effect on this LU type specifically. The model coupling in the current study adds to the uncertainty, because the (non-natural) grassland area in GLOBIOM-Brazil had to be divided into areal demands for planted pasture and rangeland in PLUC. Planted pasture and rangeland having different allocation rules, the relative distribution into the two demands determines the locations of changes.

### 4.2. *Limitations and Future Directions*

Although we have quantified the effects of model coupling, it remains unquantified how each aspect of the coupled model contributed to the found performance improvement compared to GLOBIOM-Brazil. For example, what fraction of the improvement is caused by the joined model structure of PLUC—or even by the order of allocation or the individual suitability factors within this overall model structure? Or what fraction of the improvement is caused by the initial LU map of PLUC inserted into the coupled model—or even by individual aspects of this map, such as the resolution, the different LU patterns and different LU-class areas compared to GLOBIOM-Brazil (Figure 1)?

Such questions could be analyzed by means of a sensitivity analysis, for example, doing an additional model run in which both models have the same initial LU map and comparing the difference in performance with the run in which the initial LU maps differ. Similarly to other model collaborations (e.g., [1,12]), we did not devise a common initial LU map for GLOBIOM-Brazil and PLUC in this paper, because of challenges related to the temporal mismatch between the starting years of the models (Figure 2), the models being tied to these starting years through their calibration [14,19], and the ambiguousness that would be involved in downscaling spatially (GLOBIOM-Brazil to PLUC) or thematically (PLUC to GLOBIOM-Brazil) when using one of the existing LU maps for the other model. Therefore, such a sensitivity analysis could not be performed here. We tried to address this limitation by computing the RMSE for the total areas of the LU classes in 2015 (for a large part determined by the initial map) as well as for the projected change in the areas of LU classes between 2007 and 2015 (mainly determined by the model structure) of all land-use classes in the validation. This analysis indicates that the coupled model performed better not only because of the initial LU map, but also because of model structure, but, again, without the ability to quantify the precise contributions of both aspects.

Whereas the coupled model covered the whole of Brazil, the validation was performed for the state of Mato Grosso only. On the one hand, we do not see this as a problem, as Mato Grosso is large (~10% of the total area of Brazil), and its LU is highly dynamic. On the other hand, the validation may not have been representative for LU classes with a relatively small share in Mato Grosso, such as planted forest and sugarcane. Moreover, model performance may be different in regions of Brazil with different characteristics, e.g., more densely populated regions, or other biomes. As the validation outcome depends on the observational data too, results may also differ for regions with a higher or lower data availability [1].

In the validation, the observational data were classified remote sensing images. a classification is almost never error-free. This implies that at some locations, the errors assigned to one of the models, may have been errors in the classification, leading to an underestimation of model accuracy. In future work, this limitation may be addressed by taking the uncertainty associated with the observations into account in the validation, as demonstrated by Verstegen et al. [41].

The validation was performed for a period of eight years. This period may be too short to see the effect of some of the conceptual differences between GLOBIOM-Brazil and PLUC. For example, GLOBIOM-Brazil models the re-growth of forest explicitly, while PLUC does not. This is aspect potentially projects locations of forest more accurately in GLOBIOM-Brazil compared to PLUC in the long term. Yet, to demonstrate the effect of this potentially better system description by GLOBIOM-Brazil, the validation period of eight years is too short. As another example, GLOBIOM-Brazil explicitly includes the legislation of the Soy moratorium [30], valid from 2008, and Forest code [17], in its adapted form valid from 2012. Especially for the Forest Code, the establishment may have been too late in our validation period for the model rules to shape the

results. Harmonized time series of LU maps are still scarce. a time series of LU maps with a larger coverage and longer time frame would allow validation for a larger area and a period longer than eight years.

The validation dataset does not distinguish natural from planted forest. Because natural forest had the major share in the aggregated forest class, we connected the results of the validation of the forest class mainly to natural forest. However, the projections towards 2030 showed some differences in the locations of planted forest between the two models, also in Mato Grosso. Thereby, the conclusions drawn for natural forest could have been influenced by the model differences regarding planted forest. Another limitation resulting from this is that no validation could be performed for planted forest.

Projecting changes in pasture proved to be hard, both with a single and a combined LUC model. In Section 4.1, we pointed to the fact that algorithms that extract information from remote sensing images have difficulties to distinguish grazed from natural grassland. Recent work by Parente et al. [42] on pasture mapping in Brazil provides a promising input for future work to improve the initial maps and calibration of the LUC models for the pasture class(es) and thereby, expectantly, the single and combined model performances.

Although PLUC has been used as a deterministic model before (e.g., [32]), it is originally a stochastic model [20]. Therefore, also for the Brazilian version of the model that is used here, the weights of the suitability factors and the order of allocation were originally calibrated as stochastic parameters [14], i.e., not a single value is found in the calibration but a distribution of values. In the current paper, PLUC is used deterministically, taking the medians of the calibrated parameter distributions (see Section 2.2). That means that only a single fixed order of allocation of the LU classes is used. As a result, certain transitions, for example planted forest to cropland, are not possible, whereas they would be possible in the stochastic version of PLUC. For future work, it would be interesting to couple the stochastic version of PLUC to GLOBIOM-Brazil, especially for studying the effect of having different orders of allocation in different Monte Carlo samples on coupled-model performance.

An additional advantage of a stochastic coupled model would be that maps of LU-class probabilities are obtained, allowing the computation of additional validation measures, such as the relative operating characteristic (ROC). Pontius and Schneider [43] propose to apply the ROC on the suitability map of a LUC model. This approach gives a good representation of model validity for the two-class statistical LUC model they use, but it would not reflect the validity of the coupled model, because 1) it has passive LU classes that do not have a suitability map and can thus not be accounted for by the ROC, and 2) for the active land use classes the division in macro regions and the order of allocation of the land use classes play a critical role, resulting in an allocation pattern that does not follow the suitability values. In contrast, LU-class probabilities resulting from a stochastic model run do reflect the LU allocation pattern under varying quantities of land-use change (see e.g., [14]).

Overall, model collaboration improved model performance for our case study. In addition to the evident benefits of model collaboration, the process has certain limitations. a 100% consistency between different models is very hard to achieve because of principal differences in model ontologies, semantics, structure and assumptions underlying the processes. The role of ontologies and semantics is especially high for hard linking because of the representation of the same processes in different models. Due to the differences in models' categories of LU, model comparison and validation could only be done for broad LU classes in our study, which has the drawback of losing information about the LU classes that needed to be merged during harmonization. Furthermore, the wish to maximize consistency between the models might come at a price of decreased resolution or increased computational load. In our study, certain patterns of LUC observed on PLUC's 5 km by 5 km resolution disappear when converted to GLOBIOM-Brazil's 0.5 by 0.5 decimal degree resolution. On the other hand, since understanding of LUC drivers and change in space and time is incomplete, the highest spatial resolution does not always give the most accurate model results [44].

## 5. Conclusions

In this study, we aimed to evaluate the differences between a harmonized, coupled land use change (LUC) model and a single LUC model, and to quantify the performance of both models when compared to independent observational data. We coupled the partial equilibrium (PE) model GLOBIOM-Brazil to the demand-driven spatially explicit model PLUC and projected LUC between 2007 and 2030 for Brazil. Our first research question was: What are the differences between land-use patterns produced by the coupled model and GLOBIOM-Brazil and what do these differences tell us about the models? The highest variations between the coupled model and GLOBIOM-Brazil could be observed in states Mato Grosso and Pará which are frontiers of agricultural expansion [1,21]. Furthermore, variations in model projections along the border of two biomes: Amazon and Cerrado. The projected changes in cropland, pasture, and sugarcane demonstrated greater consistency between the two models than the changes in natural forest and natural non-forest land. In terms of spatial patterns, the changes projected by the coupled model were concentrated, as a result of the cellular automata approach of PLUC, while the changes projected by GLOBIOM-Brazil were scattered, as a result of the transition-matrix approach to allocation and the lower level of detail about drivers of location of change.

Our second research question was: for which land use classes, if any, does the coupled model produce better results than GLOBIOM-Brazil when being validated against independent observational data? To answer this question, projections of change between 2007 and 2015 were validated for the state of Mato Grosso. The coupled model proved to project LUC more accurately than GLOBIOM-Brazil for four out of five land-use classes. Considering total LU (combined effect of LUC allocation and initial LU map) in 2015, the reduction in the RMSE was 57% for natural forest, 43% for sugarcane, 25% for natural non-forest land, and 10% for cropland. Only for pasture, the RMSE of the total LU was 9% higher. Considering the allocation of LUC, the coupled model performed better for all LU classes. The reduction in the root mean squared error (RSME) obtained by model coupling, was 80% for natural forest, 69% for pasture, 55% for sugarcane, 32% for natural non-forest land, and 31% for cropland. This result may be an incentive to address the current challenges that exist in model collaboration in general, and model coupling specifically [11]. An important question for future research is to what extent these performance improvements were the results of the combination of the two model structures and to what extent of the two initial LU maps.

Reasons for a better performance of the coupled model were considered to be aspects of the demand-driven model like inclusion of more and finer spatially explicit information about drivers of location of change, the path-dependence effect in the allocation of change through the cellular automata approach, and a more accurate initial LU map. Finally, the multi-scale effect obtained by the coupling better represents the multiple decision-making levels that play a role in the LUC system.

**Author Contributions:** Conceptualization, O.S., G.C. and J.A.V.; methodology, O.S.; validation, O.S. and J.A.V.; writing—original draft preparation, O.S. and J.A.V.; writing—review and editing O.S., G.C. and J.A.V.; visualization, O.S. and J.A.V.; supervision, G.C. and J.A.V.; funding acquisition, G.C. and J.A.V. All authors have read and agreed to the published version of the manuscript.

**Funding:** Research visits between the University of Münster (WWU) and National Institute for Space Research (INPE) for the preparation of this work were funded through the FAPESP/WWU SPRINT Program, Grant 2016/50495-4. The publication fees were covered by the Open Access Publication Fund of the University of Münster.

**Acknowledgments:** We greatly appreciate the feedback and suggestions from the special issue editors and two anonymous reviewers on earlier versions of this manuscript that have significantly improved our work. We would like to thank Aline Soterroni for helping us with the GLOBIOM-Brazil model. We are grateful for the grant we received through the FAPESP/WWU SPRINT Program (Grant 2016/50495-4). Finally, we acknowledge the support from the Open Access Publication Fund of the University of Münster.

**Conflicts of Interest:** The authors declare no conflict of interest. The funders had no role in the design of the study; in the collection, analyses, or interpretation of data; in the writing of the manuscript, or in the decision to publish the results.

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
