# Peer review of "Quantifying the Effect of Land Use Change Model Coupling"

_land, doi:10.3390/land9020052_

Round 1

Reviewer 1 Report

Please see attached pdf.

Author Response

Dear editors and reviewers,

Thank you very much for the quick handling of our manuscript and for finding excellent reviewers. We have worked hard, trying to accommodate all comments. The main changes compared to the original version of the manuscript are: 1) the addition of an analysis of the initial maps of PLUC (used in the coupled model) and GLOBIOM-Brazil, 2) the addition of the composite operator of Pontius and Cheuk (2006) to our existing validation metrics, 3) a more consistent use of terms in the text and colors in the figures, 4) an explicit listing of how PLUC and GLOBIOM-Brazil contribute to the coupled model, and 5) a change in the order of the result section, placing the validation before the projections.

In the attachment, we have copied every comment of the reviewers and provided a response, detailing if, how and where we have addressed the comment. The responses are preceded by the word ‘response’ and made blue and italic to allow easy recognition.

We believe that the manuscript has been improved considerably. We hope that you will reconsider our manuscript for publication in Land.

Yours sincerely,

Judith Verstegen, on behalf of the authors

Reviewer 2 Report

The submitted manuscript “Quantifying the effect of land use change model coupling” describes the results of experiments coupling two models GLOBIOM-Brazil and PLUC. Given challenges in working with the PLUC outputs and scale of operation, the authors compare GLOBIOM-Brazil to a coupled (GLOBIOM-Brazil – PLUC) model. The models are compared to observed time-series data 2007-2015 and to scenarios generated out to 2030. Coupling of land use models is of contemporary importance to the land use and land system science communities at the moment both in terms of scale of coupling (e.g., trying to link models that drive land use and land cover change like integrated assessment models with local and regional scale models that are designed to model the spatial allocation of land use change with greater accuracy) as well as coupling of land use and land cover change models with models of natural processes (see the following paper that brings researchers from each of these communities together https://www.earth-syst-dynam.net/9/895/2018/).

Model coupling and comparison is a difficult endeavour and the authors should be commended for their efforts. Having had an email exchange with one of the authors about their manuscript submission I was excited to conduct the review as one of the special issue editors. While the other editor will make a decision in isolation about this manuscript, I see it as requiring major revisions and additional peer review. Given the speed at which the journal requests revisions I have attached the submitted pdf with comments to be addressed throughout and describe the overarching major revisions requiring attention below. Highlighted sections in the attached document are some wording/English issues requiring attention. Please respond to both the comments provided below and those given in the pdf upon resubmission. Thank you.

Wording and storyline. The text requires improvement not only in the use of English throughout, but also additional effort to ensure that the word choice is appropriate and the context within which the idea/word choice resides is correctly conveyed. For example, on line 99 the manuscript includes the following text: “The results of the coupled model are upscaled to the scale and compared to the results of GLOBIOM-Brazil.” Not only does ‘upscaled to the scale’ sound strange but it causes the reader to reread aspects of the paper to find out what the scale of the models are. Another example is on line 50 where the paper mentions “real integration” without any clarification about what constitutes a real integration since you can capture feedbacks with both loose and tight coupling depending on the alignment of processes between models. With respect to the storyline, there authors try to carve out gaps in model coupling and harmonization but focus primarily on citing reference [10]. There are ample examples of harmonization, coupling, and model comparisons that exist in the literature that could and probably should be brought forward to the reader. In doing so, perhaps then you could make the argument that the presented research demonstrates and example where all three of these components are explicitly addressed. However, I would argue that most coupling papers address the harmonization and comparison pieces implicitly and are doing what reference [10] labels as model collaboration.

Structural changes to the results section. The results presents scenarios prior to presenting the results of the validation process. Without understanding the outcome of the validation process the reader has no idea how credible, likely, or how insightful the results of scenario may be. In addition to reorganizing, in the results, and elsewhere, there is a lack of quantitative data backing up general claims about the comparisons being made and justification for model comparison choices. For example, why is 50 kHa used as the threshold between large and small differences? How are hotspots defined? How do we know what major expansion from the coupled model is when it’s higher than 100 kHa per cell? Does it match peak historical expansion or is it greater than any historical expansion?

Revisiting research questions. The authors detail the following two research questions near the end of the introduction:

(1) What are the differences between land-use patterns produced by GLOBIOM-Brazil and the coupled model and what do these differences tell us about the models? (2) For which land cover classes, if any, does the coupled model produce better results than GLOBIOM-Brazil individually when being validated against independent observational data?

However, they do not revisit these research questions in the manuscript and illustrate how their results explicitly answer those questions. Doing so would tie the text together, generate a stronger storyline, and aid the authors in identifying the contributions to science that their manuscript provides, which also could be strengthened.

Results and Discussion. Based on RMSE and NSE statistics it looks like forest has the greatest variance between your models. With respect to RMSE, the coupled model shows the greatest improvement in forest and not for natural non-forest lands (Figure 4). Given these outcomes, it is difficult to interpret the first paragraph of Section 4.1 and how you have proven that “… model coupling may be generalizable to other models and case study areas.”. Please revise this paragraph and better link the interpretation and broader impacts to the results section.

The discussion section lays out a lot of content without first situating the results of the presented research in the broader context of existing literature to demonstrate the broader impacts of the presented research. It also does not immediately link the results to research questions to discuss how the questions have been answered and what their contribution to science provides. Please make these amendments before bringing in links to other’s work (e.g., [1] or discussing challenges with parameter uncertainty [14] but not actually informing the reader of the kinds or types of parameters for which the uncertainty exists.

Please consolidate and take a stance on the use of NSE in the paper. The authors note that the statistic is used in hydrological research and it’s potential utility for land use and land cover change modelling but then state at the end of the paper that it is likely more appropriate for natural science and land use and land cover change models simulate human-environment interactions that are “by nature” (also a poor choice given that you’re differentiating between human and natural process models and science) are “much more erratic and unprecitable than natural processes [7]”. Now you’ve just argued that the statistic is likely not useful in the context that you have just presented it in, which suggests that it shouldn’t be presented. Either remove the statistic (since you’ve argued it’s not useful) or prove the utility of the statistic over other typical map comparison approaches (again, lots of literature on this available from Pontius at Clark University).

If the initial land use map is a large driver of differences in the models then you can manufacture similar land use maps and see how the models compare so that you have some quantitative evidence about how much influence the initial maps actually have rather than qualitatively hypothesizing about the influence. Other ideas about this provided in the paper.

Reviewer Conclusions

I remain enthusiastic about the presented manuscript and hope that the authors will continue to revise and improve the communication of what is likely a substantial amount of time and effort. However, there are substantial conceptual, structural, and simple modifications required prior to the paper being considered for publication. The tone of the comments are only due to time constraints and are not meant to be personal in anyway. If they come off blunt, please accept my apologies. I look forward to seeing a revised manuscript.

Author Response

(The authors gave the same response as above.)

Reviewer 3 Report

This manuscript describes the ‘collaboration’ of two land use models for a Brazilian case study, analyses the accuracy gains made by model coupling and draws conclusions about the reasons for these accuracy gains in a complex region like the one selected here. I enjoyed reading this and think the authors have produced a careful, informative and useful study that deserves to be published. Coupling and comparison of models is a challenging task and relatively few papers analyse the process in the detail given here, so this makes a definite contribution. My comments mainly relate to increasing/clarifying this contribution, as I think the authors could make more of this work with a little extra effort. My comments don’t relate to the previous review, as I haven’t seen the reviewer comments or author responses.

Overall, some more general interpretation of the findings would be very welcome, and would help elevate this above a simple model comparison exercise. The authors do quietly draw some seemingly general conclusions (e.g. in the very last paragraph of the manuscript), but these are generally tied to the models being considered and often don’t go much beyond restating the results. To me it seems that a major purpose of a study like this should be to draw useful conclusions for other modellers, and indeed I think the authors will be able to do this. Points that stood out to me included the value of more accurate input data at higher resolution, the danger of fixed transition rules, the value of including policies and realistic responses to them (including in this case e.g. logging in protected areas), and spatial autocorrelation of some sort in patterns of land use change.

I’m sure the authors could add detail to these, and also perhaps discuss how generally applicable or important such lessons are. For example, this case study includes an agricultural frontier in the Amazon, and so touches on a major issue of land use change which neither model deals with very well – can the authors say anything about how can we improve our modelling of these processes?  

A linked point is that the analysis of actual changes during the simulations (as opposed to input data effects) is relatively lacking in detail, and more comparison with on-the-ground change (at least as perceived through the remote sensing data) would be very helpful. Some reasons for divergence between ‘observed’ and modelled change is given, mainly in the discussion, but discussion of the scope for accurate identification and modelling of these reasons would be useful.

Specific comments:

Abstract, lines 13-14: it’s debatable that an evaluation as described hasn’t been done before, largely because the statement is vague – edit or delete?

Line 17 – are-were

Line 40: delete ‘aiming’?

Line 56: missing space ‘practicein’

Line 62: missing apostrophe after authors

Line 72-73: a couple of typos in this sentence make it unclear

Lines 156-157: This order may be worth coming back to later – should it be a fixed property, and could changing it improve the results?

Line 162: I don’t think the term ‘hub’ has been introduced before this.

Line 186: ‘which’ instead of ‘what’

Section 2.2: It might be helpful in this section to have a table comparing the allocation methods in the two models.

Lines 183-192: the mismatch in input data is well described but is unfortunate. It seems as though more could be done on this, perhaps by upscaling the PLUC map for GLOBIOM, or applying the GLOBIOM input at higher resolution (i.e. assigning all cells within each GLOBIOM cell to the same land use). This is obviously not ideal from a modelling point of view but would do more to isolate the data and model components of the differences. The argument that input data are inherent parts of the model is a little unconvincing.

Line 216-217: ‘you practically get the coupled model’ could be better phrased.

Line 226: delete ‘s’ on models

Line 274: ‘is’ should be ‘giving’ or similar

Line 305: Incomplete sentence

Section 3.2: Perhaps mention here the ease with which pasture area can be underestimated in remote-sensed data? Also, some headings might be useful here as the same statistics are given for different cases (2007-2015, 2015), which is confusing at times.

Line 545: Write out the name or method of ref. 37?

Line 560 onwards: I’m not convinced ‘sufficient’ is a useful term here, or that the percentage threshold is useful. This is acknowledged, but it may be better to replace this with some reflection on model purpose and what might constitute sufficiency or acceptability in that context.

Lines 572-576: This very short paragraph deserves more space (possibly at the expense of detailed discussion of some of the other results). In this application, deforestation is surely a key process, and one which land use models should be seeking to represent well. Isn’t the failure of both models to manage this important? And does the comparison reveal reasons for the failure? (presumably at least that the relationships included are not enough).

Lines 599-607: This is a particularly interesting paragraph, and seems the sort of discussion from which more general conclusions can be drawn for modelling.

Line 613: ‘there was’

Author Response

Dear reviewer,

Thank you for your careful review and suggestions. They were very helpful to improve our manuscript. Please find a detailed response to each of your suggestions in the attachment.
